# Fish Oil Enriched n-3 Polyunsaturated Fatty Acids Improve Ketogenic Low-Carbohydrate/High-Fat Diet-Caused Dyslipidemia, Excessive Fat Accumulation, and Weight Control in Rats

**DOI:** 10.3390/nu14091796

**Published:** 2022-04-25

**Authors:** Shing-Hwa Liu, Yu-Xuan Chen, Huei-Ping Tzeng, Meng-Tsan Chiang

**Affiliations:** 1Institute of Toxicology, College of Medicine, National Taiwan University, Taipei 10051, Taiwan; shinghwaliu@ntu.edu.tw (S.-H.L.); hptzeng811@gmail.com (H.-P.T.); 2Department of Pediatrics, College of Medicine and Hospital, National Taiwan University, Taipei 10051, Taiwan; 3Department of Medical Research, China Medical University Hospital, China Medical University, Taichung 40402, Taiwan; 4Department of Food Science, National Taiwan Ocean University, Keelung 20224, Taiwan; a0441082@gmail.com

**Keywords:** low-carbohydrate/high-fat diet, fish oil, body weight loss, leptin, lipid metabolism

## Abstract

Low-carbohydrate and high-fat diets have been used for body weight (BW) control, but their adverse effects on lipid profiles have raised concern. Fish oil (FO), rich in omega-3 polyunsaturated fatty acids, has profound effects on lipid metabolism. We hypothesized that FO supplementation might improve the lipid metabolic disturbance elicited by low-carbohydrate and high-fat diets. Male SD rats were randomized into normal control diet (NC), high-fat diet (HF), and low-carbohydrate/high-fat diet (LC) groups in experiment 1, and NC, LC, LC + 5% FO (5CF), and LC + 10% FO diet (10CF) groups in experiment 2. The experimental duration was 11 weeks. In the LC group, a ketotic state was induced, and food intake was decreased; however, it did not result in BW loss compared to either the HF or NC groups. In the 5CF group, rats lost significant BW. Dyslipidemia, perirenal and epididymal fat accumulation, hepatic steatosis, and increases in triglyceride and plasma leptin levels were observed in the LC group but were attenuated by FO supplementation. These findings suggest that a ketogenic low-carbohydrate/high-fat diet with no favorable effect on body weight causes visceral and liver lipid accumulation. FO supplementation not only aids in body weight control but also improves lipid metabolism in low-carbohydrate/high-fat diet-fed rats.

## 1. Introduction

Dietary strategies that restrict carbohydrates have been used in clinical practice for decades. There are several types of carbohydrate-restricted diets, some of which restrict carbohydrates to very low levels, without restricting fat or total calories, to induce ketosis. For example, a classic ketogenic diet usually consists of a 4:1~3:1 ratio of fats to carbohydrates and protein, with 90% of the calories from fats [1]. There has been abundant evidence suggesting that ketogenic diets, apart from their benefits for refractory epilepsy and cancer treatment, also have beneficial effects on body weight (BW) loss and glycemic control in obese subjects and diabetic patients [2,3,4]. However, the impact of ketogenic diets may be controversial, both in rodents and humans [5]. The majority of recent studies in human subjects clearly demonstrate that physiological ketosis induced by a low carbohydrate/high fat diet ameliorates blood lipid profiles, improves metabolic syndrome, and reduces cardiovascular (CV) risk factors, such as by lowering blood triglycerides (TG), reducing total cholesterol (TC), increasing HDL-C, decreasing systolic/diastolic blood pressure, or lowering blood glucose [6,7,8]. Even a moderate low-carbohydrate diet, containing 25% total daily energy (TDE) from carbohydrates, has positive effects on weight loss and levels of HDL-C, LDL-C, and TC [9]. However, the effects of ketogenic diets in rodents differ from the observations in humans and are mostly reported to be involved with a reduction in body weight but worsening of lipid profiles, such as a reduction in HDL-C levels and an increase in LDL-C, TC, and TG levels [5]. Despite the wide use of low-carbohydrate/high-fat diets and ketogenic diets in many fields, there are some studies raising concerns about their adverse side effects in humans, such as impaired fetal growth, kidney stones, hyperlipidemia, and CV risks [10]. Moreover, numerous studies in rodents consistently suggest that ketogenic diets may lead to glucose intolerance, insulin resistance, fat accumulation, and hepatic steatosis [11,12,13,14].

Marine fish oil (FO), rich in long-chain omega-3 polyunsaturated fatty acids (n-3 PUFAs), notably eicosapentaenoic acid (EPA; 20:5 n-3) and docosahexaenoic acid (DHA; 22:6 n-3), has demonstrated numerous beneficial effects on human health, such as a reduction in the risk of cardiovascular disease (CVD), the improvement of metabolic syndromes, as well as the treatment of obesity [15,16]. These health benefits of FO may be attributed to the anti-inflammatory and hypolipidemic properties of n-3 PUFAs [17,18]. Accumulating evidence, particularly in animal models, indicates that n-3 PUFAs are believed to play a role in suppressing appetite and calorie intake by regulating adipokines such as leptin, improving insulin sensitivity, reducing plasma TG, decreasing fat mass, increasing lean mass by stimulating fat oxidation and energy expenditure, and limiting hepatic steatosis [19,20].

Given that controlling saturated fatty acid intake is crucial for preventing significant LDL-C increase and for achieving improved CV health on a low-carbohydrate/high-fat diet [4], and given FO has anti-obesity potential and can improve the metabolic profile, we hypothesized that FO supplementation might potentiate or improve the effects of low-carbohydrate/high-fat diets on BW control and lipid metabolism. To our knowledge, few studies have directly compared diets with different degrees of carbohydrate restriction in rats. Accordingly, we first fed rats with three different diets: a low-carbohydrate/high-fat diet (LC group; carbohydrates 25% of total daily energy (TDE), fats 60% of TDE), a moderate-carbohydrate and moderate-fat diet (HF group; carbohydrates ~50% of TDE, fats ~33% of TDE) as a mimic of the Western diet, and standard chow as a normal control diet (NC group; carbohydrates ~70% of TDE, fats ~11% of TDE) to observe their effects on body weight control and lipid profiles after 11 weeks of feeding. Then, we partially substituted the lard of the low-carbohydrate/high-fat diet for 5% FO (5CF group) and 10% FO (10CF group), and further observed whether the FO-substituted low-carbohydrate/high-fat diets were more effective than the low-carbohydrate/high-fat diet. Surprisingly, our study showed that the low-carbohydrate/high-fat diet did not result in BW loss compared to the HF and NC groups, although it did induce moderate ketosis. Rats fed with the 5CF diet, but not with 10CF diet, lost significant BW starting from week 6. FO supplementation attenuated low-carbohydrate/high-fat diet-caused dyslipidemia, perirenal and epididymal fat accumulation, TG increase, and hepatic steatosis. A significant increase in leptin caused by the low-carbohydrate/high-fat diet was counteracted by FO, as well, suggesting a role for leptin in the beneficial effect of FO on lipid metabolism.

## 2. Materials and Methods

### 2.1. Animals and Diets

All procedures were approved by the Animal House Management Committee of the National Taiwan Ocean University and conducted in accordance with the guidelines for the care and use of laboratory animals. For all studies, male Sprague-Dawley (SD) rats, at the age of 5 weeks, were obtained from BioLASCO Taiwan Co., Ltd. (Taipei, Taiwan) and individually housed in stainless steel cages and maintained under controlled temperature (22–24 °C), with a relative humidity of 40–60% and on a 12 h light-dark lighting cycle. After 1 week of acclimation with rodent chow (Laboratory Rodent Diet 5001, PMI Feed, Inc., St. Louis, MO, USA), rats were weighed and randomly divided into different groups. In brief, rats were ranked in order of their body weight and then were allocated from high body weight to low body weight into different groups in a “forward-backward” movement to ensure that there was no significant difference in the average initial body weight between groups.

In experiment 1, rats were assigned to three groups (*n* = 7 per group) varying in dietary carbohydrates and fats: (1) Normal control diet group (NC group): rats maintained ad libitum access to standard chow (soybean oil 2%, lard 3%, corn starch ~65%), which contained about 70% TDE from carbohydrates and 11% TDE from fats; (2) High-fat diet group (HF group): rats were fed a moderate-carbohydrate and moderate-fat diet (soybean oil 2%, lard 15%, corn starch ~53%), which mirrored the Western diet containing about 50% TDE from carbohydrates and 33% from fats; and (3) Low-carbohydrate/high-fat diet group (LC group): rats were fed a low-carbohydrate/high-fat diet (soybean oil 2%, lard 35%, corn starch 33%), a ketogenic diet from which 25% of the calories were derived from carbohydrates and 60% from fats (Table 1). In experiment 2, rats were divided into 4 groups (*n* = 8 per group), including NC, LC, LC + 5% FO (5CF), and LC + 10% FO (10CF) groups (Table 1). Protein intake was held constant (casein 20% in mass) in all groups. The sources of fat in these experimental diets were soybean oil and lard, such that the diet included both saturated and unsaturated long-chain fatty acids. The composition of fatty acids in FO was described previously [21]. All rats were allowed ad libitum access to their diet; food intake was measured 3 times a week, and BW was recorded once a week for 11 weeks. At the end of the experiments, after 12 h fasting, rats were euthanized under anesthesia (70% CO_2_/30% O_2_); blood samples were collected from the abdominal aorta into heparin-contained tubes; and liver, intestine, gastrocnemius muscle, soleus muscle, and perirenal and epididymal adipose tissues were immediately excised, weighed, and stored at −80 °C until further use. The decision on the number of animals used was based on our preliminary experiments and a previous study [21].

### 2.2. Analysis of Ketosis

Analysis of blood β-hydroxybutyrate is believed to be a more precise method of assessing the level of ketosis in rats compared to the measurement of urine acetone [22]. After 11 weeks on their respective diets, blood was collected and centrifuged, and plasma samples were stored at −80 °C until analysis. Quantification of plasma β-hydroxybutyrate was determined by a β-hydroxybutyrate colorimetric assay kit (BioVision Incorporated, Milpitas, CA, USA) and measured absorbance at 450 nm (VersaMax microplate, Molecular Device, San Jose, CA, USA).

### 2.3. Measurement of Blood Lipids, Circulating Cytokines, and Metabolic Parameters

Plasma triglycerides (TG), total cholesterol (TC), alanine aminotransferase (ALT), and aspartate aminotransferase (AST) were analyzed using commercially available kits (Randox Laboratories Ltd., Antrim, UK) according to the manufacturer’s instructions. Plasma TNF-α, IL-6, and leptin were measured using ELISA kits for rats (R&D system, Inc., Minneapolis, MN, USA).

### 2.4. Assessment of Liver Histology

Fresh liver tissues were harvested, fixed with 10% formalin, dehydrated in 70–100% ethanol, and embedded in paraffin. The paraffin blocks were then sliced into 4-µm thick sections, which were used for HE staining. Histopathological alterations in liver tissues were assessed in ten microscopic fields at ×400 magnification using a light microscope (U-LH100HG, BX53, Olympus Corporation, Tokyo, Japan) and quantified using Image-Pro Plus 6.0 (Media Cybernetics, Inc. Rockville, MD, USA).

### 2.5. Detection of Liver Lipid Content

The extraction and isolation of TG and cholesterol from livers were performed using a modified Folch’s protocol [23]. In brief, ∼0.2 g of liver tissue was homogenized in 2:1 chloroform–methanol, and total lipids were extracted. The lipids were then mixed with triton X-100 and treated by vacuum concentrator (SC 110, Savant Instruments Inc., Woonsocket, RI, USA) to remove solvents. Liver TG and TC content were then determined spectrophotometrically at 500 nm using enzymatic colorimetric kits (Randox Laboratories Ltd., Antrim, UK). Briefly, 10 μL of the liver lipid extract or standard solution was added to the kit reagent buffer (1 mL). The mixtures were incubated at 37 °C for 5 min and then measured using a spectrophotometer at 500 nm wavelength (UV/VIS-7800, JASCO International Co., Ltd., Tokyo, Japan). The equation used is as follows:

Liver TC or TG content (mg/dL) = (Es-blank)/(Estd-blank) × 200. Es is the absorbance of the sample, Estd is the absorbance of the standard, and 200 is the concentration of the standard solution.

### 2.6. Data Management and Statistical Analysis

All analyses were carried out in triplicate by well-trained personnel as per the instructions and supervised by senior personnel. The experimental results and notebook checking were verified by the Lab PI on a regular basis to ensure the accuracy and reliability of the obtained data. Quality control samples were used in the analytical methods for analytical precision. All results are presented as means ± SD. For the statistical comparison between the dietary groups, ANOVA was performed with a subsequent Tukey post hoc test using GraphPad Prism 6 (GraphPad Software, La Jolla, CA, USA). *p* value < 0.05 was set as the level of significance. Different superscript letters indicate statistical significance.

## 3. Results

### 3.1. Ketogenic Low-Carbohydrate and High-Fat Diet Has No Effect on Body Weight Control but Leads to Proinflammatory State, Dyslipidemia, Visceral Fat Deposition, and Hepatic Steatosis

Of note, the circulating β-hydroxybutyrate levels had approximately a 2-fold increase in the LC group after 11 weeks of the diet, indicating a ketotic state in rats of the LC group (Figure 1). As shown in Table 2, the body weight and the body weight gain in the LC group were 564.6 ± 38.4 and 400.6 ± 34.4 vs. 525.3 ± 32.9 and 363.0 ± 35.5 g in the NC group, showing no significant difference in body weight change between the LC and NC groups, despite the fact that food intake was significantly lower in the LC group compared to the NC group. In the HF group, body weight and body weight gain were significantly higher than in the NC group, in agreement with the well-known obesity-inducing effect of the Western high-fat diet; however, the increased body weight was not different compared to the LC group (Table 2). The daily energy intake levels of the NC and LC groups were not different, whereas rats in the LC group ate a small but significantly lower level of calories per day compared to the HF group (Table 2). These results suggest that the ketosis induction in the LC group had no effect on preventing body weight gain compared to the NC group and caused no additional body weight increase compared to the HF group.

As shown in Table 3, after 11 weeks, the plasma TNF-α levels in rats of the LC group were higher than those in the NC group but significantly lower than those in the HF group, whereas no change in interleukin-6 (IL-6) was observed among the groups. In addition, there was a significant increase in plasma leptin in the LC group compared to both the NC and HF groups. Interestingly, rats in the LC group had lower plasma levels of TC and TG than those in the NC group.

We then examined whether there were any metabolic impacts on visceral adipose tissues in the LC group. Perirenal adipose weight and epididymal adipose weight were increased in both the LC and HF groups compared to the NC group (Table 3), and the TG levels in perirenal and epididymal adipose tissue were also significantly increased in both the LC and HF groups (Figure 2). The increase in visceral adipose mass and TG content caused by the low-carbohydrate/high-fat diet was as high as that caused by the HF diet (Table 3 and Figure 2). No difference in weight or length of the small intestine was observed among these three groups (Table 3).

To further investigate whether the dyslipidemia was related to metabolic changes in the liver, the fat vacuoles, as well as hepatic TC and TG levels, were determined as a measure of hepatic steatosis. The area of fat vacuoles was increased more than two-fold in the LC group (Figure 3A), along with an increase in hepatic TC and TG levels (Figure 3B); both of these factors showed no significant difference as compared to the HF group. Moreover, plasma ALT and AST showed no change among the groups (Table 3). Taken together, the disturbance of lipid metabolism by the ketogenic low-carbohydrate/high-fat diet was characterized by evidence of visceral fat deposition in the perirenal and epididymal regions, ectopic fat accumulation in the liver, and increased TG and TC content in the liver. The extent of the metabolic disorder in the LC group was not more severe than that in the HF group, although there was a higher lard content in the composition of the low-carbohydrate/high-fat diet.

### 3.2. Fish Oil (FO) Substitution Ameliorates the Ketogenic Low-Carbohydrate and High-Fat Diet-Caused Metabolic Disorder

We first confirmed that feeding a low-carbohydrate and high-fat diet that contained 5% or 10% FO in rats (the 5CF and 10 CF groups) could also induce ketosis in rats, as in the LC group (Figure 4). Unlike in the LC group, rats in the 5CF group showed decreased food intake and daily energy intake, along with decreased body weight starting from week 6 compared to the LC group (Table 4). However, no such finding was observed in the 10CF group. Despite the fact that the TG-lowering effect in the LC group was not further potentiated by FO supplementation, both the 5% and 10% FO enhanced low-carbohydrate/high-fat diets mediated a reduction in plasma TC levels (Table 4). In addition, the visceral fat accumulation and TG content in the perirenal and epididymal locations in the LC group were also reversed by FO supplementation (Table 4 and Figure 5). No weight changes in gastrocnemius muscle or soleus muscle were observed among these four groups (Table 4). We next evaluated whether the hepatic lipid profile was altered in response to FO supplementation. There was only a small change in liver weight in both the 5CF and 10CF groups compared to both the NC and LC groups (Table 4); however, hepatic TG levels were significantly decreased (Figure 5C) and hepatic steatosis was completely prevented in rats of both the 5CF and 10CF groups compared to the LC group (Figure 6). Finally, we observed that the increased leptin levels in the LC group were capable of being reversed by FO supplementation (Table 4).

## 4. Discussion

The anti-obesity potential of FO or n-3PUFAs has been studied during the past few years [19]. The main purpose of the present study was to investigate whether FO would assist with weight loss when administered as a supplement during a low-carbohydrate/high-fat diet designed for weight loss. We first showed that rats fed ad libitum low-carbohydrate and high-fat diets demonstrated no body weight change, despite the fact that ketosis occurred. What made matters worse was that the low-carbohydrate/high-fat diet exhibited metabolic disturbances as evidenced by dyslipidemia, visceral fat accumulation in the perirenal and epididymal regions, and ectopic fat deposition in the liver. In contrast, FO supplementation showed benefits for body weight control and lipid metabolism. Rats fed a low-carbohydrate/high-fat diet with 5% of lard replaced by FO demonstrated body weight losses starting from week 6. Visceral fat accumulation and related TG content, hepatic steatosis and related TG content, as well as worsened plasma leptin levels were either partially or completely reversed by FO supplementation.

Dietary intervention studies are diverse in their design, including differences in the macronutrient composition of the diet, the isocaloric/hypocaloric intake, the source and formulation of n-3 PUFAs, the time frames of feeding, and even the timing of initiation of the intervention [24]. It is reasonable that some induce weight loss, whereas others do not. The low-carbohydrate/high-fat diet we used in the study, which consisted of approximately a 0.7:1 ratio of fats to carbohydrates and protein, induced moderate ketosis and showed no favorable effect on body weight; however, it exhibited a lower daily calorie intake, lower plasma and liver TG content, as well as lower TNF-α levels as compared to the Western high-fat diet, which actually contained less fat in its composition. We believe that ketosis induced by the low-carbohydrate and high-fat diet might have participated in the benefits demonstrated since ketogenic diets have been reported to have appetite suppressing, TG-lowering, and anti-inflammatory effects [3,25]. We also observed that the low-carbohydrate/high-fat diet-fed rats had a higher body fat content compared to the normal control diet-fed rats; the degree of fat accumulation was similar to that seen in the high-fat diet-fed rats, which is in accordance with their deleterious metabolic effects [26,27]. This is likely a consequence of the nature of high-fat diets, leading to an increase in fat deposition in different tissues, such as the liver and the perirenal and epididymal adipose tissue, independent of caloric consumption [28].

The effects of ketogenic diets on lipid profiles are known to be controversial. In the present study, we observed that TG and TC levels were reduced in the ketogenic LC group, which is consistent with the findings reported by Garbow et al. [14] and by Holland et al. [29]. TGs are synthesized via lipogenesis in the liver and secreted as VLDL into the blood and, in turn, into peripheral tissues, including adipose tissue and muscle. Lipogenesis also takes place in adipose tissue. Lipogenesis in both the liver and adipose tissue can be stimulated by a high-carbohydrate diet [30]. In contrast, the restriction of carbohydrate intake, such as in our LC group, may reduce lipogenesis; thus, it resulted in lower plasma TG levels compared to the NC group. However, a high-fat diet may increase liver lipogenesis but decrease VLDL secretion, which increases fat accumulation in the liver. Plus, lipolysis is also increased in adipose tissues by overfeeding with SFAs, and the release of free fatty acids may subsequently stimulate lipogenesis in the liver, further contributing to liver fat accumulation [24].

Studies relating effects on body weight and energy intake with n-3 PUFA supplementation are inconsistent [31,32,33]. Our data showed that the inclusion of 5% FO in the low-carbohydrate and high-fat diet decreased energy intake along with a significant effect on body weight loss; however, 10% FO supplementation had no such effects. In addition, some studies have indicated that n-3 PUFAs play a role in reducing adipose tissue mass, such as in the epididymal and retroperitoneal locations [34,35], which is consistent with our finding that perirenal and epididymal adipose weights were significantly decreased by both the 5% and 10% FO-supplemented low-carbohydrate and high-fat diets, as compared to the LC group. It is well accepted that diets with different amounts of n-3 PUFAs, as well as different experiment durations, may provide different outcomes [36]. After 11 weeks on the diets, we found that the 5FC group saw better effects in terms of lowering energy intake and body weight gain and similar impacts on reducing lipid accumulation in adipose tissues and the liver, as compared to the 10FC group, suggesting that the 5FC group might have had a better ratio of n-6 to n-3 PUFAs to regulate appetite and body weight for the duration of the experiment [37]. However, a previous research study demonstrated that fish oil supplementation decreased total fatty acids in plasma lipids, with a decrease in total SFA and monounsaturated fatty acids (MUFAs) and n-6 PUFAs but an increase in n-3 PUFAs, as compared to the lard group in mice. They also observed that the amount of n-3 PUFAs in plasma increased roughly proportionally to the composition of the diet [38]. In a human study, an increase in the amount of EPA, DHA, total n-3 fatty acids, the DHA/arachidonic acid ratio, as well as a lower ratio of SFA and n-6/n-3 in plasma were shown after supplementation with fish oil [39]. Of note, accumulating evidence suggests that different sources of n-3 PUFAs, in addition to fish oil, have benefits on lipid metabolism as well, such as krill oil, flaxseed oil, and marine diatoms [40,41,42]. Moreover, n-6 PUFAs, such as sunflower, have been reported to reduce fat accumulation, despite their proinflammatory potential [43,44].

Multiple mechanisms by which n-3 PUFAs improve metabolic profiles and reduce body weight have been proposed. Omega-3 PUFAs may increase fatty acid oxidation in the liver, adipose tissue, and cardiac and skeletal muscle, thus limiting fat deposition in these tissues [36]. They may also reduce hepatic lipogenesis and VLDL secretion, thus limiting the delivery of fatty acids to peripheral adipose tissues [45], which may explain our observation of reduced adipose mass, reversed hepatic steatosis, and decreased TG content in adipose tissue and the liver caused by FO. Moreover, n-3 PUFAs may also stimulate skeletal muscle protein synthesis or preserve muscle mass during energy restriction [46,47]; however, in the present study, we did not see any weight changes in gastrocnemius or soleus muscles among the various groups.

Non-alcoholic fatty liver disease (NAFLD), or the accumulation of liver fat, has been recognized as the hepatic expression of metabolic syndrome and may have long-term adverse metabolic consequences [48]. NAFLD, in the beginning, is simple steatosis; however, it may progress to its more severe form of non-alcoholic steatohepatitis (NASH), which is characterized by liver inflammation and hepatocellular injury and may further progress to cirrhosis and ultimately develop into liver cancer [49]. In the present study, hepatic steatosis was observed in the LC group; it was simple, without inflammatory infiltration or cellular injury, as determined by the evidence of no elevations in plasma ALT or AST levels. A number of studies have suggested that high-fat diets with saturated fatty acids (SFAs) can lead to increased liver fat content, and PUFAs have been consistently shown to have a favorable effect on liver steatosis, regardless of body weight change [44,50]; this agrees with our findings. FO supplementation reversed low-carbohydrate/high-fat diet-mediated hepatic fat accumulation; thus, it may protect the liver from further progressing into a more severe stage of liver disease.

Moreover, n-3 PUFAs may also modulate the secretion of adipokines, such as leptin [51]. Leptin, predominantly produced and secreted from white adipose tissue, is believed to play a vital role in the regulation of energy homeostasis and is capable of suppressing appetite, promoting energy expenditure, and regulating BW [52]. It has been shown that obese individuals have high serum leptin levels [53], and hyperleptinemia is sufficient and necessary to induce leptin resistance. Evidence suggests that leptin resistance is often characterized by reduced satiety, over-consumption of nutrients, and increased total body mass, which may subsequently lead to obesity and is associated with a higher incidence of metabolic syndromes [54,55]. In this study, unfavorable levels of circulating leptin induced by the low-carbohydrate/high-fat diet were completely reversed by FO supplementation, suggesting that the anti-obesity effects of n-3 PUFAs may be through regulating leptin resistance, leading to a reduction in body weight gain. This finding is supported by other published work in the literature, both in rodents and in humans [33,56,57]. The underlying mechanisms by which FO regulates leptin-mediated BW control are worth further investigation.

## 5. Conclusions

To date, only a few rodent studies have directly compared low-carbohydrate/high-fat diets with the “standard” Western HF diet, and we observed that a moderate ketogenic low-carbohydrate/high-fat diet, despite having a greater amount of fat compared to the high-fat diet, induced less inflammation and showed no visible difference in the degree of visceral fat deposition or hepatic steatosis. FO supplementation, in addition to preventing BW gain, improved the low-carbohydrate/high-fat diet-induced metabolic disturbance of lipids. Thus, according to our findings, it is better to receive medical supervision while employing a ketogenic diet or a low-carbohydrate/high-fat diet. In addition, lowering SFA intake by the substitution of FO or n-3 PUFAs for SFAs is another solution for avoiding the potential adverse effects of ketogenic low-carbohydrate/high-fat diets. Further studies are needed to better understand the beneficial effect of FO on NAFLD, with a specific focus on the related mechanisms. Improving body composition by a decrease in body fat and an increase in/preservation of muscle mass by way of long-term FO supplementation may be another topic to explore in the future. Overall, if a low-carbohydrate/high-fat diet is adopted to manage BW, a combination of carbohydrate restriction and FO supplementation may be the more effective and safer strategy for reducing weight and improving lipid metabolic disorders. However, there are limitations to fatty acid analysis; it would be helpful to understand the differences between the 5% and 10% FO diets.

## Figures and Tables

**Figure 1 nutrients-14-01796-f001:**
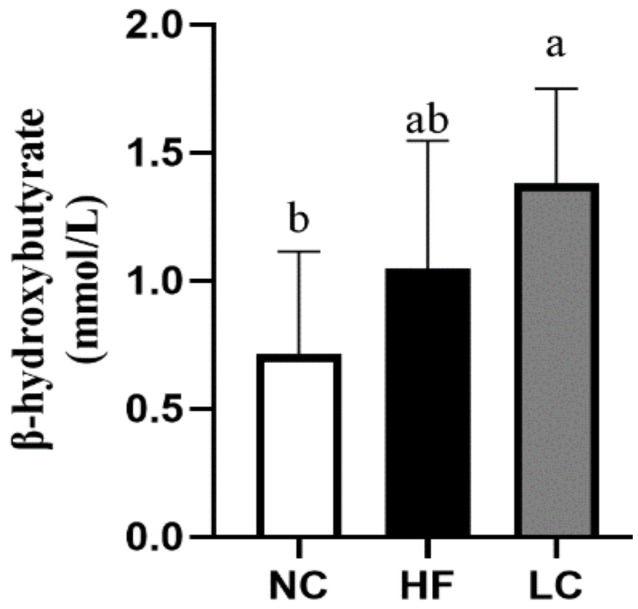
The plasma levels of β-hydroxybutyrate in rats fed with the different experimental diets for 11 weeks. Results are expressed as mean ± SD for each group (*n* = 7). Mean values with different letters are significantly different, *p* < 0.05.

**Figure 2 nutrients-14-01796-f002:**
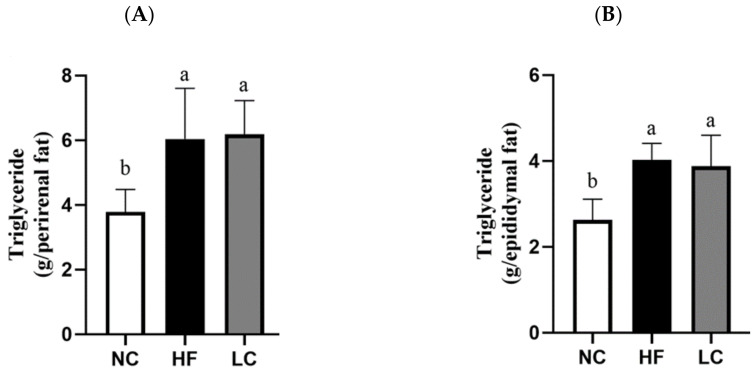
The triglyceride concentration in perirenal and epididymal adipose tissue of rats fed with different experimental diets for 11 weeks. The levels of triglyceride in (**A**) perirenal and (**B**) epididymal adipose are shown. Results are expressed as the mean ± SD for each group (*n* = 7). Values with different letters are significantly different, *p* < 0.05.

**Figure 3 nutrients-14-01796-f003:**
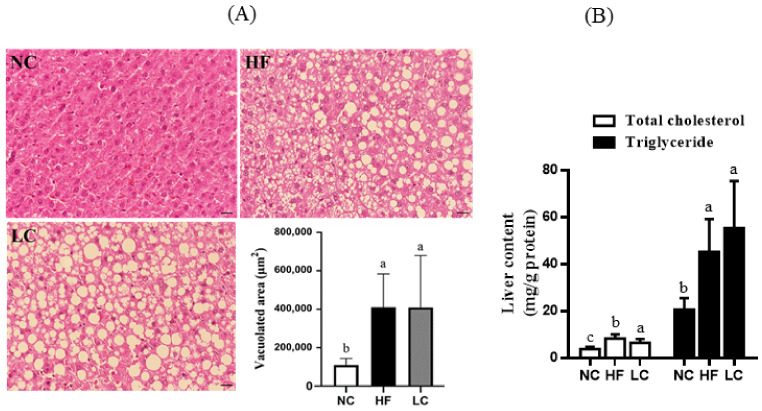
The hepatic morphology and lipid metabolites in rats fed with different experimental diets for 11 weeks. Representative hematoxylin and eosin (HE)-stained images of fat vacuoles in liver tissues are shown and quantified (**A**). The hepatic total cholesterol and triglyceride content are shown in (**B**). Results are expressed as the mean ± SD for each group (*n* = 7). Mean values with different letters are significantly different, *p* < 0.05.

**Figure 4 nutrients-14-01796-f004:**
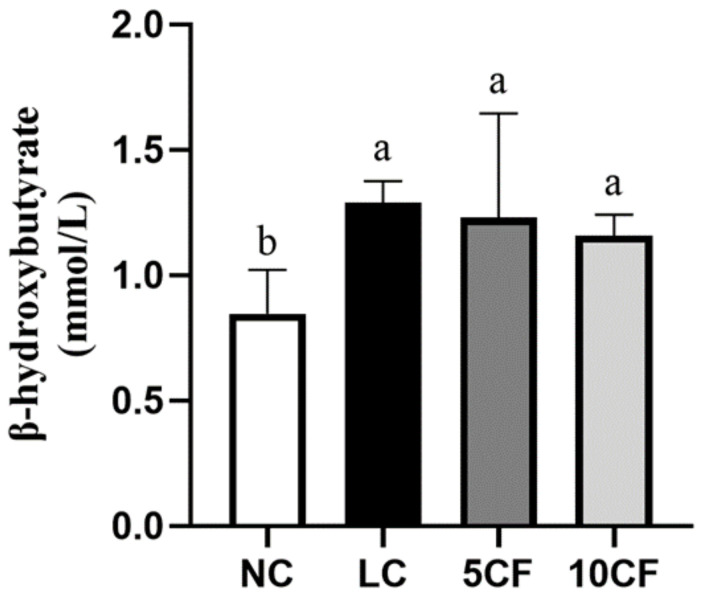
The plasma levels of β-hydroxybutyrate in rats fed with different experimental diets for 11 weeks. Results are expressed as mean ± SD for each group (*n* = 8). Mean values with different letters are significantly different, *p* < 0.05.

**Figure 5 nutrients-14-01796-f005:**
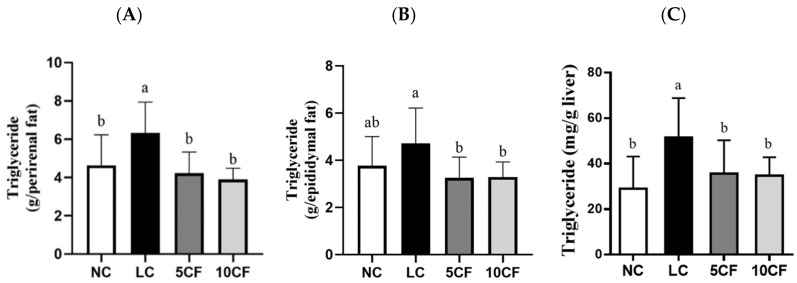
The triglyceride concentration in the perirenal and epididymal adipose tissue and in the liver of rats fed with different experimental diets for 11 weeks. The levels of triglycerides in (**A**) perirenal adipose, (**B**) epididymal adipose, and (**C**) liver were measured. Results are expressed as the mean ± SD for each group (*n* = 8). Values with different letters were significantly different, *p* < 0.05.

**Figure 6 nutrients-14-01796-f006:**
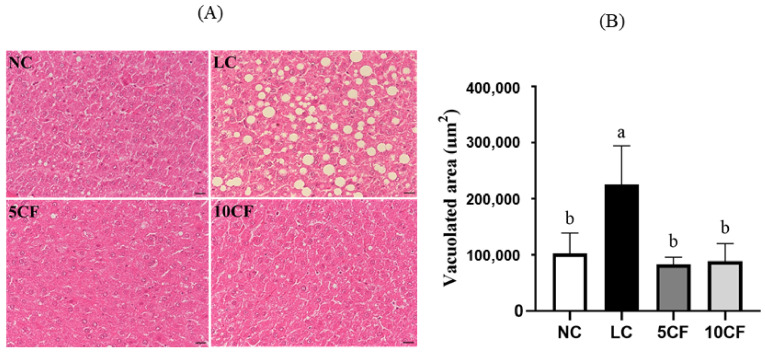
The hepatic morphological change in rats fed with different experimental diets for 11 weeks. Representative hematoxylin and eosin (HE)-stained images of fat vacuoles in liver tissues are shown (**A**) and quantified (**B**). Results are expressed as the mean ± SD for each group (*n* = 8). Mean values with different letters are significantly different, *p* < 0.05.

**Table 1 nutrients-14-01796-t001:** Experimental diet composition.

Ingredient (%)	NC	HF	LC	5CF	10CF
Corn starch	64.8	52.8	32.8	32.8	32.8
Casein	20	20	20	20	20
Soybean oil	2	2	2	2	2
Lard	3	15	35	30	25
Fish oil	0	0	0	5	10
Choline chloride	0.2	0.2	0.2	0.2	0.2
Vitamins ^1^	1	1	1	1	1
Minerals ^2^	4	4	4	4	4
Cellulose	5	5	5	5	5
Total calories (kcal/100 g)	394.2	454.2	554.2	554.2	554.2
Carbohydrate (%)	68.290	48.701	25.478	25.478	25.478
Protein (%)	20.294	17.613	14.435	14.435	14.435
Fat (%)	11.416	33.686	60.087	60.087	60.087
Total (%)	100	100	100	100	100

NC: Normal control diet (Soybean oil 2% and Lard 3%); HF: High-fat diet (Soybean oil 2% and Lard 15%); LC: High-fat, low-carbohydrate diet (Soybean oil 2% and Lard 35%); 5CF: High-fat, low-carbohydrate diet (Soybean oil 2%, Lard 30%, and fish oil 5%); 10CF: High-fat, low-carbohydrate diet (Soybean oil 2%, Lard 25%, and fish oil 10%). ^1^ AIN-93 vitamin mixture; ^2^ AIN-93 mineral mixture.

**Table 2 nutrients-14-01796-t002:** The body weight and food intake in rats fed with the different experimental diets for 11 weeks.

	NC	HF	LC
Initial body weight (g)	162.3 ± 8.76 ^a^	164.2 ± 7.80 ^a^	164.1 ± 8.12 ^a^
Final body weight (g)	525.3 ± 32.9 ^b^	591.1 ± 32.5 ^a^	564.6 ± 38.4 ^ab^
Body weight gain (g)	363.0 ± 35.5 ^b^	426.9 ± 32.6 ^a^	400.6 ± 34.4 ^ab^
Food intake (g/day)	28.1 ± 1.21 ^a^	24.6 ± 1.53 ^b^	19.1 ± 1.05 ^c^
Calorie intake (kcal/day)	110.0 ± 4.82 ^ab^	114.5 ± 6.4 ^a^	107.5 ± 6.85 ^b^

NC: Normal control diet (Soybean oil 2% and Lard 3%); HF: High-fat diet (Soybean oil 2% and Lard 15%); LC: High-fat, low-carbohydrate diet (Soybean oil 2% and Lard 35%). Results are expressed as the mean ± SD for each group (*n* = 7). Different superscript letters indicate statistical significance (*p* < 0.05).

**Table 3 nutrients-14-01796-t003:** Plasma biochemical parameters and organ weights in rats fed with the different experimental diets for 11 weeks.

	NC	HF	LC
Total cholesterol (mg/dL)	93.77 ± 15.58 ^a^	78.95 ± 20.34 ^ab^	66.2 ± 13.91 ^b^
Triglyceride (mg/dL)	87.14 ± 18.51 ^a^	60.02 ± 18.27 ^b^	31.63 ± 16.98 ^c^
Leptin (ng/mL)	4.06 ± 1.39 ^a^	6.99 ± 2.65 ^ab^	8.15 ± 2.74 ^b^
TNF-α (pg/mL)	4.27 ± 1.72 ^a^	13.75 ± 4.10 ^b^	8.75 ± 2.77 ^c^
IL-6 (pg/mL)	66.57 ± 8.02 ^a^	66.93 ± 11.35 ^a^	62.65 ± 12.94 ^a^
AST (U/L)	20.3 ± 4.44 ^a^	19.5 ± 2.47 ^a^	20.3 ± 2.39 ^a^
ALT (U/L)	15.2 ± 5.20 ^a^	14.5 ± 3.92 ^a^	16.3 ± 7.13 ^a^
Liver weight (g)	17.0 ± 2.28 ^a^	16.8 ± 1.41 ^a^	15.1 ± 1.11 ^a^
Relative liver weight (g/100 g BW)	3.22 ± 0.28 ^a^	2.84 ± 0.23 ^b^	2.68 ± 0.11 ^b^
Perirenal adipose weight (g)	14.3 ± 2.62 ^b^	21.9 ± 5.75 ^a^	22.6 ± 3.65 ^a^
Epididymal adipose weight (g)	9.59 ± 1.81 ^b^	14.7 ± 1.34 ^a^	14.0 ± 2.50 ^a^
Small intestine weight (g)	7.97 ± 1.46 ^a^	8.50 ± 0.92 ^a^	8.59 ± 0.39 ^a^
Small intestine length (cm)	122.9 ± 11.4 ^a^	116.4 ± 8.32 ^a^	120.0 ± 3.27 ^a^

NC: Normal control diet (Soybean oil 2% and Lard 3%); HF: High-fat diet (Soybean oil 2% and Lard 15%); LC: High-fat, low-carbohydrate diet (Soybean oil 2% and Lard 35%). Results are expressed as means ± SD for each group (*n* = 7). Different superscript letters indicate statistical significance (*p* < 0.05).

**Table 4 nutrients-14-01796-t004:** Effects of fish oil supplementation on the body weight, food intake, plasma biochemical parameters, and tissue weights in rats fed with different experimental diets for 11 weeks.

	NC	LC	5CF	10CF
Initial body weight (g)	170.4 ± 12.4 ^a^	171.3 ± 12.3 ^a^	172.9 ± 11.1 ^a^	172.5 ± 9.5 ^a^
Mid-term body weight (g)	456.4 ± 34.8 ^ab^	465.8± 31.5 ^a^	427.7 ± 14.7 ^b^	438.9 ± 19.1 ^ab^
Final body weight (g)	555.1 ± 50.8 ^ab^	577.6 ± 52.1 ^a^	534.0 ± 23.4 ^b^	553.5 ± 28.2 ^ab^
Body weight gain (g)	384.8 ± 48.7 ^a^	406.3 ± 51.3 ^a^	361.1 ± 27.1 ^a^	381.0 ± 27.6 ^a^
Food intake (g/day)	26.7 ± 2.02 ^a^	19.2 ± 1.71 ^b^	17.1 ± 0.54 ^c^	18.2 ± 0.68 ^bc^
Calorie intake (kcal/day)	105.4 ± 6.18 ^a^	106.2 ± 6.41 ^a^	94.53 ± 5.27 ^b^	100.9 ± 5.34 ^ab^
Total cholesterol (mg/dL)	120.3 ± 30.4 ^a^	76.9 ± 23.8 ^b^	44.8 ± 20.1 ^c^	32.8 ± 11.3 ^c^
Triglyceride (mg/dL)	126.7 ± 53.4 ^a^	38.2 ± 15.7 ^b^	23.9 ± 18.6 ^b^	16.4 ± 9.65 ^b^
Leptin (ng/mL)	8.53 ± 4.18 ^b^	13.04 ± 4.63 ^a^	6.97 ± 2.73 ^b^	8.53 ± 4.18 ^b^
AST (U/L)	32.2 ± 18.5 ^a^	23.9 ± 10.6 ^a^	19.3 ± 8.79 ^a^	27.6 ± 14.9 ^a^
ALT (U/L)	24.2 ± 11.3 ^a^	16.4 ± 5.77 ^a^	17.8 ± 7.61 ^a^	17.2 ± 3.92 ^a^
Liver weight (g)	17.5 ± 3.65 ^a^	14.5 ± 1.61 ^b^	15.0 ± 1.81 ^b^	16.2 ± 0.89 ^ab^
Relative liver weight (g/100 g BW)	3.12 ± 0.41 ^a^	2.52 ± 0.18 ^c^	2.80 ± 0.27 ^bc^	2.93 ± 0.23 ^ab^
Perirenal adipose weight (g)	15.3 ± 4.60 ^b^	21.5 ± 5.09 ^a^	13.7 ± 3.22 ^b^	12.9 ± 1.3 ^b^
Epididymal adipose weight (g)	11.7 ± 3.73 ^ab^	14.7 ± 4.58 ^a^	10.3 ± 2.68 ^b^	10.4 ± 1.89 ^b^
Gastrocnemius muscle weight (g)	6.08 ± 0.63 ^a^	5.96 ± 0.37 ^a^	5.77 ± 0.55 ^a^	5.94 ± 0.44 ^a^
Soleus muscle weight (g)	0.38 ± 0.05 ^a^	0.42 ± 0.04 ^a^	0.39 ± 0.04 ^a^	0.39 ± 0.08 ^a^

NC: Normal control diet (Soybean oil 2% and Lard 3%); HF: High-fat diet (Soybean oil 2% and Lard 15%); LC: High-fat, low-carbohydrate diet (Soybean oil 2% and Lard 35%); 5CF: High-fat, low-carbohydrate diet (Soybean oil 2%, Lard 30%, and fish oil 5%); 10CF: High-fat, low-carbohydrate diet (Soybean oil 2%, Lard 25%, and fish oil 10%). Results are expressed as the mean ± SD for each group (*n* = 8). Mid-term body weights were measured after 6 weeks of diet. Mean values with different superscript letters indicate statistical significance (*p* < 0.05).

## Data Availability

The data presented in this study are available from the corresponding author upon reasonable request.

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
