# Peer review of "Fish Oil Enriched n-3 Polyunsaturated Fatty Acids Improve Ketogenic Low-Carbohydrate/High-Fat Diet-Caused Dyslipidemia, Excessive Fat Accumulation, and Weight Control in Rats"

_nutrients, 2022, doi:10.3390/nu14091796_

Round 1
Reviewer 1 Report
The article is of a very high scientific level. The subject of the research is very interesting and raises an issue important for nutrition. Paper’s layout is correct, in line with the guidelines for writing research papers. After properly conducted statistical analysis of the research results, conclusions adequate to the assumptions of the study were formulated. The discussion is a comparison of obtained results with the research of other authors in the area of ​​the issue, current positions of international literature were taken into account. The paper was written in a structured, precise, stylistically and linguistically correct way. The paper is interesting with a significant cognitive value and the possibility of using the results obtained in practical terms. The results were very clear and accurately presented in tables. Numerous charts and images add value to the article.
The section of material and methods should contain information about the exact number of rats in the different groups. The randomization process of assigning rats to groups should be described in more detail.
Author Response
-Reviewer 1
The article is of a very high scientific level. The subject of the research is very interesting and raises an issue important for nutrition. Paper’s layout is correct, in line with the guidelines for writing research papers. After properly conducted statistical analysis of the research results, conclusions adequate to the assumptions of the study were formulated. The discussion is a comparison of obtained results with the research of other authors in the area of the issue, current positions of international literature were taken into account. The paper was written in a structured, precise, stylistically and linguistically correct way. The paper is interesting with a significant cognitive value and the possibility of using the results obtained in practical terms. The results were very clear and accurately presented in tables. Numerous charts and images add value to the article.
The section of material and methods should contain information about the exact number of rats in the different groups. The randomization process of assigning rats to groups should be described in more detail.
Response: Thank you for your comments. The assignment of rats and the number of rats in each group were included in the “2.1 Animals and Diets” section, lines 96-99, 100, and 110. The corresponding paragraph has been written as follows:
All procedures were approved by the Animal House Management Committee of the National Taiwan Ocean University and conducted in accordance with the guide-lines for care and use of laboratory animals. For all studies, male Sprague-Dawley (SD) rats, at age of 5 weeks, were obtained from BioLASCO Taiwan Co., Ltd. (Taipei, Tai-wan), and individually housed in stainless steel cages and maintained under controlled temperature (22-24 °C), with relative humidity of 40%-60%, and on a 12-h light-dark lighting cycle. After 1 week of acclimation with rodent chow (Laboratory Rodent Diet 5001, PMI Feed, Inc., St. Louis, MO, USA), rats were weighted and randomly divided into different groups. In brief, rats were ranked in order of their body weights, and then were allocated from high body weight to low body weight into different groups in a “forward-backward” movement, to ensure the average of initial body weight between groups was no big difference.
In experiment 1, rats were assigned to three groups (n=7 per group) varying in dietary carbohydrates and fats: (1) Normal control diet group (NC group) …… In experiment 2, rats were divided into 4 groups (n=8 per group), including NC, LC, LC+5% FO (5CF), and LC+10% FO (10CF) groups……
Reviewer 2 Report
The manuscript by Liu et al on “Fish oil Enriched n-3 Polyunsaturated Fatty Acids Improves ketogenic Low-Carbohydrate/high-Fat Diet-Caused Dyslipidemia, Excessive Fat Accumulation and Weight Control in Rats” describes an interesting experiment, which shows the importance of fatty acid composition on dietary effects in rats.
This is an interesting experiment with results, which seem valuable. Nevertheless, I would like to raise some points.
It would be good to seem some “quantitative” comparisons with other studies, i.e. are the decreases observed for triglycerides and cholesterol in the circulation similar to previous observations in rats on ketogenic diets?
Are there associations between serum and tissue levels of TG and CHOL?
Concerning the fat components this seems like a comparison of lard and fish oil; it would be good to see fatty acid data.
It seems important to discuss, how important the selection of the fat source is, would it look different, if instead of lard a more unsaturated fat had been used, e.g. canola oil?
There seems to be no difference between the 5% and 10% fish oil groups, what could that mean?
How was the number of rats per group defined?
Specific points:
Line 44: increasing LDL-cholesterol? It seems ref 6 says differently
Line 55: demonstrated prevention of obesity by EPA and DHA seems too strong to me
Line 120: “more” than what?
Analytical method: generally, please add info on the quality control measures and indicators of analytical precision
Line 145: how works an enzymatic kit in a lipid extract? Please provide details, was it within the specifications of the Randox kit?
Figure 2 and other figures: g/perirenal fat? Is it total fat?
Author Response
-Reviewer 2
The manuscript by Liu et al on “Fish oil Enriched n-3 Polyunsaturated Fatty Acids Improves ketogenic Low-Carbohydrate/high-Fat Diet-Caused Dyslipidemia, Excessive Fat Accumulation and Weight Control in Rats” describes an interesting experiment, which shows the importance of fatty acid composition on dietary effects in rats.
This is an interesting experiment with results, which seem valuable. Nevertheless, I would like to raise some points.
- It would be good to see some “quantitative” comparisons with other studies, i.e. are the decreases observed for triglycerides and cholesterol in the circulation similar to previous observations in rats on ketogenic diets?
Response: We appreciate your comments and suggestions. The added sentences were included in “Introduction” and “Discussion” sections, lines 46-49 and 263-275. The corresponding paragraph has been written as follows:
In “Introduction” section
However, the effects of ketogenic diets in rodents differ from the observations in humans, and mostly reported to be involved in reduction of body weight but worseness of the lipid profiles, such as reduction of HDL-C levels and increase of LDL-C, TC and TG levels [5].
In “Discussion” section
The effects of ketogenic diets on lipid profiles are known to be controversial. In the present study, we observed that the TG and TC levels were reduced in LC group, which is consistent with the findings reported by Garbow et. al. [14] and by Holland et. al. [29]. ……
- Are there associations between serum and tissue levels of TG and CHOL?
Response: We appreciate your comment. Yes. A new paragraph on this discussion was included in the discussion section, lines 266-275.
TGs are synthesized via lipogenesis in the liver and secreted as VLDL into blood and in turn into peripheral tissues including adipose tissue and muscle. Lipogenesis also take place in adipose tissue. Lipogenesis in both liver and adipose tissue can be stimulated by a high carbohydrate diet [30]. In contrast, restriction of carbohydrate intake, such as in our LC group may reduce lipogenesis and resulted in lower plasma TG levels, compared to NC group. However, high fat diet may increase liver lipogenesis but decrease VLDL secretion, which increases the fat accumulation in the liver. Plus, Lipolysis is also increased in adipose tissues by overfeeding of SFA, and the release of free fatty acid may subsequently stimulate lipogenesis in the liver, further contributing the liver fat accumulation [24].
- Concerning the fat components this seems like a comparison of lard and fish oil; it would be good to see fatty acid data.
Response: We appreciate your comment. We have analyzed the component of fish oil in our previous study (Chiu et al., 2018, Ref. 21), but we didn’t check the fatty acid profile after feeding of experimental diets in the current study. It will be an interesting idea to pursue in the future. We have discussed this issue in the Discussion of this revised manuscript, lines 291-297. The corresponding paragraph has been written as follows and also shown together in the next question.
However, a previous research demonstrated that fish oil supplementation decreased total fatty acids in plasma lipids, with a decrease of total SFA and monounsaturated fatty acids (MUFAs) and n-6 PUFAs but an increase of n-3 PUFAs, compared to the lard group in mice. They also observed the amount of n-3 PUFAs in plasma increased roughly proportionally to the composition of the diet [38]. In human study, a raise of the amount of EPA and DHA, total n-3 fatty acids and DHA/arachidonic acid ratio, as well as a lower of SFA and n-6/n-3 ratio in plasma were shown after supplementation of fish oil [39].
- It seems important to discuss, how important the selection of the fat source is, would it look different, if instead of lard a more unsaturated fat had been used, e.g. canola oil?
Response: We appreciate your comment. We have discussed this issue in the Discussion, lines 297-301. Accumulating evidence suggests that different source of n-3 PUFAs, in addition to fish oil, has benefits on lipid metabolism as well, such as krill oil, flaxseed oil and marine diatoms [40-42]. Besides, n-6 PUFAs, such as sunflower, have been reported to reduce fat accumulation, despite their proinflammatory potential [43,44]. The corresponding paragraph has been written as follows:
Of note, accumulating evidence suggests that different source of n-3 PUFAs, in addition to fish oil, has benefits on lipid metabolism as well, such as krill oil, flaxseed oil and marine diatoms [40-42]. Besides, n-6 PUFAs, such as sunflower, have been reported to reduce fat accumulation, despite their proinflammatory potential [43,44].
- There seems to be no difference between the 5% and 10% fish oil groups, what could that mean?
Response: We appreciate your comment. Actually, there are some differences between 5CF and 10CF groups. It’s well acceptable that diet with different amount of n-3 PUFAs as well as different experiment duration may provide different outcomes. After 11 weeks on diets, we found that 5FC group has better effects on lowering energy intake and body weight gain and have similar impacts on reducing adipose tissue mass and liver steatosis, compared to 10FC group, suggesting 5FC group might have better ratio of PUFAs to SFAs to regulate appetite and body weight control in the duration of the experiment. Further studies on fatty acid profile between groups may give better understating the difference. We have discussed this issue in the Discussion, lines 284-290. The corresponding paragraph has been written as follows:
It’s well acceptable that diet with different amount of n-3 PUFAs as well as different experiment duration may provide different outcomes [36]. After 11 weeks on diets, we found that 5FC group has better effects on lowering energy intake and body weight gain and have similar impacts on reducing lipid accumulation in adipose tissue liver, compared to 10FC group, suggesting 5FC group might have better ratio of n-6 to n-3 PUFAs to regulate appetite and body weight in the duration of the experiment [37]. Further studies on fatty acid profile between groups might give better understating the difference.
- How was the number of rats per group defined?
Response: We appreciate your comment. In experiment 1, each group had 7 rats; and in experiment 2, each group had 8 rats. We had added the number of rats to “2.1 Animals and Diets” section.
Specific points:
(1) Line 44: increasing LDL-cholesterol? It seems ref 6 says differently.
Response: We appreciate your comment. We agreed with the reviewer on that issue. The effects of low carbohydrate/high fat diet on LDL-C are controversial. LDL-C did not change significantly in Ref 6; however, was increase in Ref 7. Given that we did not detect the expression of LDL-C in our study, we then did not put much effort on describing the controversial phenomenon. In order to make the statement simple and clear to readers, we deleted the words “and LDL-C” in this sentence.
(2) Line 55: demonstrated prevention of obesity by EPA and DHA seems too strong to me
Response: Thank you for your comment. We deleted the words “and prevention” to better express the information that the cited references [15, 16] delivered.
(3) Line 120: “more” than what?
Response: Thank you for your comment. As described in Ref 22, urinary ketone body measurement had high variation, which might be due to the differences in individual drinking behavior and renal filtration rates. They therefore suggest that analysis of serum β-hydroxybutyrate is a more precise method to assess the level of ketosis in rats. Accordingly, we added “, compared to measurement of urine acetone” to the end of the sentence.
2.2 Analysis of Ketosis
Analysis of blood β-hydroxybutyrate is believed to be a more precise method to assess the level of ketosis in rats, compared to measurement of urine acetone [22].
(4) Analytical method: generally, please add info on the quality control measures and indicators of analytical precision
Response: Thank you for your comment. All analyses were carried out in triplicate by well-trained personnel as per the instructions, and supervised by senior personnel. The experimental results and notebook checking were verified by Lab PI, on a regular basis, to ensure accuracy and reliability of obtained data. The above sentences were added to the “2.6. Data Management and Statistical Analysis” section. The corresponding paragraph has been written as follows:
2.6. Data Management and Statistical Analysis
All analyses were carried out in triplicate by well-trained personnel as per the instructions, and supervised by senior personnel. The experimental results and notebook checking were verified by Lab PI, on a regular basis, to ensure accuracy and reliability of obtained data.
(5) Line 145: how works an enzymatic kit in a lipid extract? Please provide details, was it within the specifications of the Randox kit?
Response: Thank you for your comment. Briefly, 10 μL of the liver lipid extract or standard (std) solution was added to 1 mL of kit reagent buffer. The mixtures were incubated at 37°C for 5 min and then measured using a spectrophotometer at 500 nm wavelength (UV/VIS-7800, JASCO International Co., Ltd., Tokyo, Japan). The equation used is as follows: Liver TC or TG (mg/dL) = (Es-blank)/(Estd-blank) × 200. Es is the absorbance of sample; Estd is the absorbance of standard, 200 is the concentration of standard solution. We have added these descriptions in the Methods, lines 149-154. The corresponding paragraph has been written as follows:
Briefly, 10 μL of the liver lipid extract or standard solution was added to kit reagent buffer (1mL). The mixtures were incubated at 37°C for 5 min and then measured using a spectrophotometer at 500 nm wavelength (UV/VIS-7800, JASCO International Co., Ltd., Tokyo, Japan). The equation used is as follows:
Liver TC or TG content (mg/dL) = (Es-blank)/(Estd-blank) × 200. Es is the absorbance of sample; Estd is the absorbance of standard, 200 is the concentration of standard solution.
(6) Figure 2 and other figures: g/perirenal fat? Is it total fat?
Response: Thank you for your comment. In general, the total visceral fat of male rats was estimated by measurement of the sum of the mesenteric, perirenal, and epididymal fat pads. In this study, we measured only the perirenal and epididymal fat.
Round 2
Reviewer 2 Report
Dear authors,
thank you very much for your response to the comments.
I would liketo come back to some specific points:
Number of animals: I think the important point here is: What is the basis for the decision on the used number of animals? Are there enough animals to draw firm conclusions? Which data were used to define the number.
Analytical precision: Thanks for the detailed description, but it would be good to know the observed variation of replicated analyses. Have qualitycontrol samples been included?
As you mention that fatty acid analyses would be helpful to understand the differences between the 5 and 10 % diets, I think it would be good to mention that as limitation of the study.
Author Response
Reviewer 2
thank you very much for your response to the comments.
I would like to come back to some specific points:
(1) Number of animals: I think the important point here is: What is the basis for the decision on the used number of animals? Are there enough animals to draw firm conclusions? Which data were used to define the number.
Response: We appreciate the reviewer’s comment. The decision on the used number of animals was based on our preliminary experiments and the previous study (such as ref. 21). We have added this description in the Methods of this revised manuscript according to the suggestion of reviewer.
(2) Analytical precision: Thanks for the detailed description, but it would be good to know the observed variation of replicated analyses. Have quality control samples been included?
Response: We appreciate the reviewer’s comment. We have used the quality control samples, such as method blanks, calibration blanks, and calibration curve standards, in replicated analyses for analytical precision. We have added the description for quality control samples in the Methods section of this revised manuscript.
(3) As you mention that fatty acid analyses would be helpful to understand the differences between the 5 and 10 % diets, I think it would be good to mention that as limitation of the study.
Response: We appreciate the reviewer’s comment. We have added the descriptions for this issue in the Conclusions of this revised manuscript according to the suggestion of reviewer.
There is a limitation for fatty acid analysis that would be helpful to understand the differences between the 5 and 10 % FO diets.